# Region Mutual Information Loss for Semantic Segmentation

**Shuai Zhao**[1], **Yang Wang**[2], **Zheng Yang**[3], **Deng Cai**[1,4*]

[1]State Key Lab of CAD&CG, College of Computer Science, Zhejiang University
[2]School of Artificial Intelligence and Automation, Huazhong University of Science and Technology
[3]Fabu Inc., Hangzhou, China
[4]Alibaba-Zhejiang University Joint Institute of Frontier Technologies
zhaoshuaimcc@gmail.com, wangyang_sky@hust.edu.cn, yangzheng@fabu.ai, dcai@zju.edu.cn

## Abstract

Semantic segmentation is a fundamental problem in computer vision. It is considered as a pixel-wise classification problem in practice, and most segmentation models use a pixel-wise loss as their optimization criterion. However, the pixel-wise loss ignores the dependencies between pixels in an image. Several ways to exploit the relationship between pixels have been investigated, *e.g.*, conditional random fields (CRF) and pixel affinity based methods. Nevertheless, these methods usually require additional model branches, large extra memories, or more inference time. In this paper, we develop a region mutual information (RMI) loss to model the dependencies among pixels more simply and efficiently. In contrast to the pixel-wise loss which treats the pixels as independent samples, RMI uses one pixel and its neighbour pixels to represent this pixel. Then for each pixel in an image, we get a multi-dimensional point that encodes the relationship between pixels, and the image is cast into a multi-dimensional distribution of these high-dimensional points. The prediction and ground truth thus can achieve high order consistency through maximizing the mutual information (MI) between their multi-dimensional distributions. Moreover, as the actual value of the MI is hard to calculate, we derive a lower bound of the MI and maximize the lower bound to maximize the real value of the MI. RMI only requires a few extra computational resources in the training stage, and there is no overhead during testing. Experimental results demonstrate that RMI can achieve substantial and consistent improvements in performance on PASCAL VOC 2012 and CamVid datasets. The code is available at https://github.com/ZJULearning/RMI.

## 1 Introduction

Semantic segmentation is a fundamental problem in computer vision, and its goal is to assign semantic labels to every pixel in an image. Recently, much progress has been made with powerful convolutional neural networks (*e.g.*, VGGNet [33], ResNet [14], Xception [8]) and fancy segmentation models (*e.g.*, FCN [23], PSPNet [40], SDN [11], DeepLab [5, 6, 7], ExFuse [39], EncNet [38]). These segmentation approaches treat semantic segmentation as a pixel-wise classification problem and solve it by minimizing the average pixel-wise classification loss over the image. The most commonly used pixel-wise loss for semantic segmentation is the softmax cross entropy loss:

$$\mathcal{L}_{ce}(y, p) = -\frac{1}{N} \sum_{n=1}^{N} \sum_{c=1}^{C} y_{n,c} \log(p_{n,c}), \tag{1}$$

---

[*]Deng Cai is the corresponding author

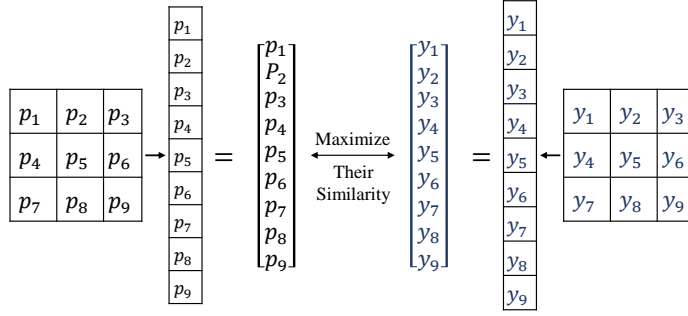

Figure 1: An image region and its corresponding multi-dimensional point. Following the same strategy, an image can be cast into a multi-dimensional distribution of many high dimensional points, which encode the relationship between pixels.

where $y \in \{0, 1\}$ is the ground truth label, $p \in [0, 1]$ is the estimated probability, $N$ denotes the number of pixels, and $C$ represents the number of object classes. Minimizing the cross entropy between $y$ and $p$ is equivalent to minimizing their relative entropy, *i.e.*, Kullback-Leibler (KL) divergence [12].

As Eq. (1) shows, the softmax cross entropy loss is calculated pixel by pixel. It ignores the relationship between pixels. However, there are strong dependencies existed among pixels in an image and these dependencies carry important information about the structure of the objects [37]. Consequently, models trained with a pixel-wise loss may struggle to identify the pixel when its visual evidence is weak or when it belongs to objects with small spatial structures [18], and the performance of the model may be limited.

Several ways to model the relationship between pixels have been investigated, *e.g.*, Conditional Random Field (CRF) based methods [5, 19, 32, 22, 41] and pixel affinity based methods [18, 21, 25]. Nevertheless, CRF usually has time-consuming iterative inference routines and is sensitive to visual appearance changes [18], while pixel affinity based methods require extra model branches to extract the pixel affinity from images or additional memories to hold the large pixel affinity matrix. Due to these factors, the top-performing models [11, 40, 6, 7, 39, 38] do not adopt these methods.

In this paper, we develop a region mutual information (RMI) loss for semantic segmentation to model the relationship between pixels more simply and efficiently. Our work is inspired by the region mutual information for medical image registration [30]. The idea of RMI is intuitive, as shown in Fig. 1, given a pixel, if we use this pixel and its 8-neighbours to represent this pixel, we get a 9-dimensional (9-D) point. For an image, we can get many 9-D points, and the image is cast into a multi-dimensional (multivariate) distribution of these 9-D points. Each 9-D point also represents a small $3 \times 3$ region, and the relationship between pixels is encoded in these 9-D points.

After we get the two multi-dimensional distributions of the ground truth and the prediction given by the segmentation model, our purpose is to maximize their similarity. The mutual information (MI) is a natural information-theoretic measure of the independence of random variables [17]. It is also widely used as a similarity measure in the field of medical image registration [36, 24, 28, 30, 35]. Thus the prediction and ground truth can achieve higher order consistency through maximizing the MI between their multi-dimensional distributions than only using a pixel-wise loss. However, the pixels in an image are dependent, which makes the multi-dimensional distribution of the image hard to analysis. This means calculating the actual value of the MI between such two undetermined distributions becomes infeasible. So we derive a lower bound of the MI, then we can maximize this lower bound to maximize the real value of the MI between two distributions.

We adopt a downsampling strategy before constructing the multi-dimensional distributions of the prediction and ground truth. The goal is to reduce memory consumption, thus RMI only requires a few additional computational resources during training. In this way, it can be effortlessly incorporated into any existing segmentation frameworks without any changes to the base model. RMI also has no extra inference steps during testing.

Experimental results demonstrate that the RMI loss can achieve substantial and consistent improvements in performance on PASCAL VOC 2012 [10] and CamVid [4] datasets. We also empirically compare our method with some existing pixel relationship modeling techniques [19, 20] on these two datasets, where RMI outperforms the others.

## 2 Entropy and mutual information

Let $X$ be a discrete random variable with alphabet $\mathcal{X}$, its probability mass function (PMF) is $p(x), x \in \mathcal{X}$. For convenience, we denote $p(x)$ and $p(y)$ refer to two different random variables, and they are actually two different PMFs, $p_X(x)$ and $p_Y(y)$ respectively. The entropy of $X$ is defined as [9]:

$$H(X) = -\sum_{x \in \mathcal{X}} p(x) \log p(x). \tag{2}$$

The entropy is a measure of the uncertainty of a random variable [9]. Then the joint entropy $H(X, Y)$ and conditional entropy $H(Y|X)$ of a pair of discrete random variables $(X, Y)$ with a joint distribution $p(x, y)$ and a conditional distribution $p(y|x)$ are defined as:

$$H(X, Y) = -\sum_{x \in \mathcal{X}} \sum_{y \in \mathcal{Y}} p(x, y) \log p(x, y), \tag{3}$$

$$H(Y|X) = -\sum_{x \in \mathcal{X}} \sum_{y \in \mathcal{Y}} p(x, y) \log p(y|x). \tag{4}$$

Indeed, the entropy of a pair of random variables is the entropy of one plus the conditional entropy of the other [9]:

$$H(X, Y) = H(X) + H(Y|X). \tag{5}$$

We now introduce mutual information, which is a measure of the amount of information that $X$ and $Y$ contain about each other [9]. It is defined as:

$$I(X; Y) = \sum_{x \in \mathcal{X}} \sum_{y \in \mathcal{Y}} p(x, y) \log \frac{p(x, y)}{p(x)p(y)}. \tag{6}$$

The equation (6) suggests that $I(X; Y)$ is a very natural measure for dependence [17]. It can also be considered as the reduction in the uncertainty of $X$ due to the knowledge of $Y$, and vice versa:

$$I(X; Y) = H(X) - H(X|Y) = H(Y) - H(Y|X). \tag{7}$$

Similar definitions of continuous random variables can be found in [9, Chap.8].

## 3 Methodology

As shown in Fig. 1, we can get two multivariate random variables, $\boldsymbol{P} = [p_1, p_2, \ldots, p_d]^T$ and $\boldsymbol{Y} = [y_1, y_2, \ldots, y_d]^T$, where $\boldsymbol{P} \in \mathbb{R}^d$ is the predicted probability, $\boldsymbol{Y} \in \mathbb{R}^d$ denotes the ground truth, $p_i$ is in $[0, 1]$, and $y_i$ is 0 or 1. If the square region size in Fig. 1 is $\mathcal{R} \times \mathcal{R}$, then $d = \mathcal{R} \times \mathcal{R}$. The probability density functions (PDF) of $\boldsymbol{P}$ and $\boldsymbol{Y}$ are $f(p)$ and $f(y)$ respectively, their joint PDF is $f(y, p)$. The distribution of $\boldsymbol{P}$ can also be considered as the joint distribution of $p_1, p_2, \ldots, p_d$, and it means $f(p) = f(p_1, p_2, \ldots, p_d)$. The mutual information $I(\boldsymbol{Y}; \boldsymbol{P})$ is defined as [9]:

$$I(\boldsymbol{Y}; \boldsymbol{P}) = \int_{\mathcal{Y}} \int_{\mathcal{P}} f(y, p) \log \frac{f(y, p)}{f(y)f(p)} \mathrm{d}y \mathrm{d}p, \tag{8}$$

where $\mathcal{Y}$ and $\mathcal{P}$ are the support sets of $\boldsymbol{Y}$ and $\boldsymbol{P}$ respectively. Our purpose is to maximize $I(\boldsymbol{Y}; \boldsymbol{P})$ to achieve high order consistency between $\boldsymbol{Y}$ and $\boldsymbol{P}$.

To get the mutual information, one straightforward way is to find out the above PDFs. However, the random variables $p_1, p_2, \ldots, p_d$ are dependent as pixels in an image are dependent. This makes their joint density function $f(p)$ hard to analyze. In [30], Russakoff *et al.* demonstrated that, for grayscale images, $\boldsymbol{Y}$ and $\boldsymbol{P}$ are normally distributed when $\mathcal{R}$ is large enough, and it is well supported by the $m$-dependence variable concept proposed by Hoeffding *et al.* [16]. Nevertheless, in our situation, we find that when $\boldsymbol{Y}$ and $\boldsymbol{P}$ are normally distributed, the side length $\mathcal{R}$ becomes very large, *e.g.*, $\mathcal{R} \geq 30$. Then the dimensions $d$ is larger than 900, and memory consumption becomes extremely large. Thus the implementation of this method is unrealistic. Due to these factors, we derive a lower bound of $I(\boldsymbol{Y}; \boldsymbol{P})$ and maxmize the lower bound to maximize the actual value of $I(\boldsymbol{Y}; \boldsymbol{P})$.

### 3.1 A lower bound of mutual information

From Eq. (7), we have $I(\boldsymbol{Y}; \boldsymbol{P}) = H(\boldsymbol{Y}) - H(\boldsymbol{Y}|\boldsymbol{P})$. At the same time, we know that a normal distribution maximizes the entropy over all distributions with the same covariace [9, Theorem 8.6.5]. And the entropy of a normal distribution with a covariance matrix $\boldsymbol{\Sigma} \in \mathbb{R}^{d \times d}$ is $\frac{1}{2} \log \left( (2\pi e)^d \det(\boldsymbol{\Sigma}) \right)$, where $\det(\cdot)$ is the determinant of the matrix. We can thus get a lower bound of the mutual information $I(\boldsymbol{Y}; \boldsymbol{P})$:

$$
\begin{aligned}
I(\boldsymbol{Y}; \boldsymbol{P}) &= H(\boldsymbol{Y}) - H(\boldsymbol{Y}|\boldsymbol{P}) \\
&\geq H(\boldsymbol{Y}) - \frac{1}{2} \log \left( (2\pi e)^d \det(\boldsymbol{\Sigma}_{Y|P}) \right).
\end{aligned}
\tag{9}
$$

where $\boldsymbol{\Sigma}_{Y|P}$ is the posterior covariance matrix of $\boldsymbol{Y}$, given $\boldsymbol{P}$. It is a symmetric positive semidefinite matrix. This lower bound is also discussed in [29]. Following the commonly used cross entropy loss (Eq. (1)), we ignore the constant terms which are not related to the parameters in the model. Then we get a simplified lower bound to maximize:

$$
I_l(\boldsymbol{Y}; \boldsymbol{P}) = -\frac{1}{2} \log \left( \det(\boldsymbol{\Sigma}_{Y|P}) \right).
\tag{10}
$$

### 3.2 An approximation of posterior variance

At this point, the key problem turns out to find out the posterior covariance matrix $\boldsymbol{\Sigma}_{Y|P}$. However, we cannot get the exact $\boldsymbol{\Sigma}_{Y|P}$ because we do not know PDFs of $\boldsymbol{Y}$ and $\boldsymbol{P}$ or their dependence. Fortunately, Triantafyllopoulos *et al.* [34] have already given an approximation of the posterior variance under a certain assumption in Bayesian inference.

Suppose we need to estimate $\boldsymbol{Y}$, given $\boldsymbol{P}$. $\mathbb{E}(\boldsymbol{Y})$ is the mean vector of $\boldsymbol{Y}$ (also $\boldsymbol{\mu}_y$), $\mathrm{Var}(\boldsymbol{Y})$ is the variance matrix of $\boldsymbol{Y}$ (also $\boldsymbol{\Sigma}_Y$), and $\mathrm{Cov}(\boldsymbol{Y}, \boldsymbol{P})$ is the covariance matrix of $\boldsymbol{Y}$ and $\boldsymbol{P}$. Triantafyllopoulos *et al.* [34] denote the notation $\boldsymbol{Y} \perp_2 \boldsymbol{P}$ to indicate that $\boldsymbol{Y}$ and $\boldsymbol{P}$ are second order independent, *i.e.*, $\mathbb{E}(\boldsymbol{Y}|\boldsymbol{P} = p) = \mathbb{E}(\boldsymbol{Y})$ and $\mathrm{Var}(\boldsymbol{Y}|\boldsymbol{P} = p) = \mathrm{Var}(\boldsymbol{Y})$, for any value $p$ of $\boldsymbol{P}$. The second order independence is a weaker constraint than strict mutual independence. Furthermore, the regression matrix $A_{yp}$ of $\boldsymbol{Y}$ on $\boldsymbol{P}$ is introduced, which is well known as $A_{yp} = \mathrm{Cov}(\boldsymbol{Y}, \boldsymbol{P})\boldsymbol{\Sigma}_P^{-1}$. It is easy to find that $\boldsymbol{Y} - A_{yp}\boldsymbol{P}$ and $\boldsymbol{P}$ are uncorrelated by calculating their linear correlation coefficient. To obtain the approxiamation of the posterior covariance matrix $\mathrm{Var}(\boldsymbol{Y}|\boldsymbol{P} = p)$, Triantafyllopoulos *et al.* [34] assume that

$$
(\boldsymbol{Y} - A_{yp}\boldsymbol{P}) \perp_2 \boldsymbol{P}.
\tag{11}
$$

This assumption means that $\mathrm{Var}(\boldsymbol{Y} - A_{yp}\boldsymbol{P}|\boldsymbol{P} = p)$ does not depend on the value $p$ of $\boldsymbol{P}$. Following the property of covariance matrix and the definition of the second order independence, we can get:

$$
\begin{aligned}
\mathrm{Var}(\boldsymbol{Y}|\boldsymbol{P} = p) &= \mathrm{Var}(\boldsymbol{Y} - A_{yp}\boldsymbol{P}|\boldsymbol{P} = p) \\
&= \mathrm{Var}(\boldsymbol{Y} - A_{yp}\boldsymbol{P}) \\
&= \boldsymbol{\Sigma}_Y - \mathrm{Cov}(\boldsymbol{Y}, \boldsymbol{P})(\boldsymbol{\Sigma}_P^{-1})^T \mathrm{Cov}(\boldsymbol{Y}, \boldsymbol{P})^T.
\end{aligned}
\tag{12}
$$

**Theorem 3.1.** *Consider random vectors $\boldsymbol{Y}$ and $\boldsymbol{P}$ as above. Under quadratic loss, $\boldsymbol{\mu}_y + A_{yp}(\boldsymbol{P} - \boldsymbol{\mu}_p)$ is the Bayes linear estimator if and only if $(\boldsymbol{Y} - A_{yp}\boldsymbol{P}) \perp_2 \boldsymbol{P}$.*

The assumption Eq. (11) is supported by theorem 3.1 [34, Theorem 1]. The theorem suggests that if one accepts the assumptions of Bayes linear optimality, the assumption (11) must be employed. Under the assumption (11), we can get the linear minimum mean squared error (MMSE) estimator. This also demonstrates that the difference between the approximation of posterior variance (Eq. (12)) and the real posterior variance is restricted in a certain range. Otherwise, $\mathbb{E}(\boldsymbol{Y}|\boldsymbol{P}) = \boldsymbol{\mu}_y + A_{yp}(\boldsymbol{P} - \boldsymbol{\mu}_p)$ cannot be the linear MMSE estimator. Theoretic proof of theorem 3.1 and some examples are given in [34]. Now, we can get an approximation of Eq. (10):

$$
I_l(\boldsymbol{Y}; \boldsymbol{P}) \approx -\frac{1}{2} \log \left( \det \left( \boldsymbol{\Sigma}_Y - \mathrm{Cov}(\boldsymbol{Y}, \boldsymbol{P})(\boldsymbol{\Sigma}_P^{-1})^T \mathrm{Cov}(\boldsymbol{Y}, \boldsymbol{P})^T \right) \right).
\tag{13}
$$

For brevity, we set $\boldsymbol{M} = \boldsymbol{\Sigma}_Y - \mathrm{Cov}(\boldsymbol{Y}, \boldsymbol{P})(\boldsymbol{\Sigma}_P^{-1})^T \mathrm{Cov}(\boldsymbol{Y}, \boldsymbol{P})^T$, where $\boldsymbol{M} \in \mathbb{R}^{d \times d}$ and it is a positive semidefinite matrix cause it is a covariance matrix of $(\boldsymbol{Y} - A_{yp}\boldsymbol{P})$.

# 4 Implementation details

In this section, we will discuss some devil details about implementing RMI in practice.

**Downsampling**   As shown in Fig. 1, we choose pixels in a square region of size $\mathcal{R} \times \mathcal{R}$ to construct a multi-dimensional distribution. If $\mathcal{R} = 3$, this leads to 9 times memory consumption. For a float tensor with shape $[16, 513, 513, 21]$, its original memory usage is about 0.33GB, and this usage turns to be about $9 \times 0.33 = 2.97$GB with RMI. This also means more floating-point operations. We cannot afford such a large computational resource cost, so we downsample the ground truth and predicted probability to save resources with little sacrifice of the performance.

**Normalization**   From the Eq. (13), we can get $\log\big(\det(\boldsymbol{M})\big) = \sum_{i=1}^{d} \log \lambda_i$, where $\lambda$ is the eigenvalues of $\boldsymbol{M}$. It is easy to see that the magnitude of the $I_l(\boldsymbol{Y}; \boldsymbol{P})$ is very likely related to the number of eigenvalues of $\boldsymbol{M}$. To normalize the value of $I_l(\boldsymbol{Y}; \boldsymbol{P})$, we divide it by $d$:

$$I_l(\boldsymbol{Y}; \boldsymbol{P}) \approx -\frac{1}{2d} \log\big(\det(\boldsymbol{M})\big). \tag{14}$$

**Underflow issue**   The magnitude of the probabilities given by softmax or sigmoid operations may be very small. Meanwhile, the number of points may be very large, *e.g.*, there are about $263\,000$ points in a label map with size $513 \times 513$. Therefore, when we use the formula $\mathrm{Cov}(\boldsymbol{Y}, \boldsymbol{Y}) = \mathbb{E}\big((\boldsymbol{Y} - \boldsymbol{\mu}_y)(\boldsymbol{Y} - \boldsymbol{\mu}_y)^T\big)$ to calculate the covariance matrix, some entries in the matrix will have extremely small values, and we may encounter underflow issues when calculating the determinant of the covariance matrix in Eq. (14). So we rewrite the Eq. (14) as [27, Page59]:

$$I_l(\boldsymbol{Y}; \boldsymbol{P}) \approx -\frac{1}{2d} \mathrm{Tr}\big(\log(\boldsymbol{M})\big), \tag{15}$$

where $\mathrm{Tr}(\cdot)$ is the trace of a matrix. Furthermore, $\boldsymbol{M}$ is a symmetric positive semidefinite matrix. In practice, we add a small positve constant to the diagnoal entries of $\boldsymbol{M}$, then we get $\boldsymbol{M} = \boldsymbol{M} + \xi\boldsymbol{I}$, where $\xi$ to be $1e-6$ in practice. This has little effect on the optima of the system, but we can accelerate the computation of Eq. (15) by performing Cholesky decomposition of $\boldsymbol{M}$, as $\boldsymbol{M}$ a symmetric positive definite matrix now. This is already supported by PyTorch [26] and Tensorflow [1]. Moreover, double-precision floating-point numbers are used to ensure computational accuracy when calculating the value of RMI. It is also worth noting that $\log\big(\det(\boldsymbol{M})\big)$ is concave when $\boldsymbol{M}$ is positive definite [3], which makes RMI easy to optimize.

**Overall objective function**   The overall objective function used for training the model is:

$$\mathcal{L}_{all}(y, p) = \lambda\mathcal{L}_{ce}(y, p) + (1 - \lambda)\frac{1}{B}\sum_{b=1}^{B}\sum_{c=1}^{C}\big(-I_l^{b,c}(\boldsymbol{Y}; \boldsymbol{P})\big), \tag{16}$$

where $\lambda \in [0, 1]$ is a weight factor, $\mathcal{L}_{ce}(y, p)$ is the normal cross entropy loss between $y$ and $p$, $B$ denotes the number of images in a mini-batch, and the maximization of RMI is cast as a minimization problem. The role of the normal cross entropy loss is a measure of the similarity between the pixel intensity of two images, and RMI can be considered as a measure of the structural similarity between two images. Following the structural similarity (SSIM) index [37], the importance of pixel similarity and structural similarity is considered equally, so we simply set $\lambda = 0.5$.

We adopt the sigmoid operation rather than softmax operation to get predicted probabilities. This is because RMI is calculated channel-wise, we do not want to introduce interference between channels. Experimental results demonstrate the performance of models trained with softmax and sigmoid cross entropy losses is roughly the same.

# 5 Experiments

## 5.1 Experimental setup

**Base model.** We choose the DeepLabv3 [6] and DeepLabv3+ [7] as our base models. DeepLabv3+ model adds a decoder module to DeepLabv3 model to refine the segmentation results. The backbone

network is ResNet-101 [14], and the fisrt one 7×7 convolutional layer is replaced with three 3×3 convolutional layers [6, 7, 15]. The backbone network is *only* pretrained on ImageNet [31][2].

**Datasets.** We evaluate our method on two dataset, PASCAL VOC 2012 [10] and CamVid [4] datasets. PASCAL VOC 2012 dataset contains 1 464 (*train*), 1 449 (*val*), and 1 456 (*test*) images. It contains 20 foreground object classes and one background class. CamVid dataset is a street scene dataset, which contains 367 for training, 101 for validation, and 233 for testing. We use the resized version of CamVid dataset provided by SegNet [2]. It contains 11 object classes and an unlabelled class, and the image size is 480×360.

**Learning rate and training steps.** The warm up learning rate strategy introduced in [15] and the poly learning rate policy are adopted. If the initial learning rate is $lr$ and the current iteration step is $iter$, for the first *slow_iters* steps, the learning rate is $lr \times \frac{iter}{slow\_iters}$, and for the rest of the steps, the learning rate is $lr \times (1 - \frac{iter-slow\_iters}{max\_iter-slow\_iters})^{power}$ with *power* = 0.9. Here *max_iter* is the maximum training steps. For PASCAL VOC 2012 dataset, the model is trained on the *trainaug* [13] set which contains 10 582 images, *max_iter* is about $30K$, $lr = 0.007$, and *slow_iters* = $1.5K$. For CamVid dataset, we train the model on the train and validation sets, *max_iter* is about $6K$, $lr = 0.025$, and *slow_iters* = $300$.

**Crop size and output stride.** During training, the batch size is always 16. The crop size is 513 and 479 for PASCAL VOC 2012 and CamVid datasets respectively. The output stride, which is the ratio of input image spatial resolution to final output resolution, is always *16* during training and inference. When calculating the loss, we upscale the logits (output of the model before softmax or sigmoid operations) back to the input image resolution rather than downsampling it [6].

**Data augmentation.** We apply data augmentation by randomly scaling the input images and randomly left-right flipping during training. The random scale is in $[0.5, 0.75, 1.0, 1.25, 1.50, 1.75, 2.0]$ on PASCAL VOC 2012 dataset and $0.75 \sim 1.25$ on CamVid dataset. Then we standardly normalized the data so that they will have zero mean and one variance.

**Inference strategy and evaluation metric.** During inference, we use the original image as the input of the model, and *no special inference strategy* is applied. The evaluation metric is the mean intersection-over-union (mIoU) score. The rest settings are the same as DeepLabv3+ [7].

## 5.2   Methods of comparison

We compare RMI with CRF [19] and the affinity field loss [18] experimentally. Both two methods also try to model the relationship between pixels in an image to get better segmentation results.

**CRF.** CRF [19] tries to model the relationship between pixels and enforce the predictions of pixels which have similar visual appearances to be more consistent. Following DeepLabv2 [5], we use it as a post-processing step. The negative logarithm of the predicted probability is used as unary potential. We reproduce the CRF according to its offical code[3] and python wrapper[4]. Some important parameters $\theta_\alpha$, $\theta_\beta$, and $\theta_\gamma$ in [19, Eq.(3)] are set to be 30, 13, and 3 respectively as recommended. Other settings are the default. As CRF has time-consuming inference steps, it is unacceptable in some situations, *e.g.*, a real-time application scenario.

**Affinity field loss.** The affinity field loss [18] exploits the relationship between pairs of pixels. For paired neighbour pixels which belong to the same class, the loss imposes a grouping force on the pixel pairs to make their predictions more consistent. As for the paired neighbour pixels which belong to different classes, affinity field loss imposes a separating force on them to make their predictions more inconsistent. The affinity field loss adopts an 8-neighbour strategy, so it requires a large memory to hold $8\times$ ground truth label and predicted probabilities when calculating its value. To overcome this problem, Ke *et al.* [18] downsample the label when calculating the loss. However, this may hurt the performance of the model [6]. We reproduce the loss according to its official implementation[5]. When choosing the neighbour pixels in a square region, we set the region size to be $3 \times 3$ as suggested.

Table 1: Evaluation of RMI on PASCAL VOC 2012 *val* set. The CE and BCE are softmax and sigmoid cross entropy loss respectively. The data of method CE* is the experimental data with similar settings (output stride = *16*, only ImageNet pretraining, and no special inference strategies) to ours reported in [6, 7]. CRF-X means that we do inference with X iteration steps when employing CRF. The Inf. Time is the average inference time per image during testing. As CRF is used as a post-processing step, the additional time with various base models is the same. Here we apply RMI after downsampling the prediction and ground truth through average pooling, whose kernel size is $4 \times 4$ and stride is 4. The square region size is $3 \times 3$, which means the dimension of the multi-dimensional points is 9.

(a) **ResNet101-DeepLabv3**

|  | Method | Inf. Time | mIoU (%) |
|---|---|---|---|
|  | CE* [6] | unknown | 77.21 |
|  | CE | 0.12 s | 77.14 |
|  | BCE | 0.12 s | 77.09 |
|  | CE & CRF-1 | 0.39 s | 78.32 |
| DeepLabv3 | CE & CRF-5 | 0.71 s | 78.40 |
|  | CE & CRF-10 | **1.11** s | 78.28 |
|  | Affinty | 0.12 s | 76.24 |
|  | RMI | 0.12 s | **78.71** |

(b) **ResNet101-DeepLabv3+**

|  | Method | mIoU (%) |
|---|---|---|
|  | CE* [7] | 78.85 |
|  | CE | 78.17 |
|  | BCE | 77.41 |
|  | CE & CRF-1 | 78.90 |
| DeepLabv3+ | CE & CRF-5 | 78.75 |
|  | CE & CRF-10 | 78.60 |
|  | Affinty | 77.09 |
|  | RMI | **79.66** |

Table 2: Per-class results on PASCAL VOC 2012 *test* set.

| Method |  | backg. | aero. | bike | bird | boat | bottle | bus | car | cat | chair | cow | d.table | dog | horse | mbike | person | p.plant | sheep | sofa | train | tv | mIoU (%) |
|---|---|---|---|---|---|---|---|---|---|---|---|---|---|---|---|---|---|---|---|---|---|---|---|
| DeepLabv3 | CE | 94.10 | 79.58 | 41.16 | 84.67 | 67.68 | 75.09 | 87.69 | 87.40 | 92.07 | 39.66 | 83.39 | 69.68 | 86.67 | 87.10 | 86.92 | 84.39 | 65.69 | 86.66 | 57.39 | 75.28 | 75.94 | 76.58 |
|  | CRF-5 | **94.60** | 84.28 | **41.83** | 88.00 | 68.81 | 76.56 | 87.69 | 87.90 | **93.79** | 40.35 | 84.92 | **70.26** | **88.84** | **89.22** | 87.39 | 85.73 | 67.45 | 87.95 | 58.80 | 75.31 | 77.38 | 77.96 |
|  | RMI | 94.57 | **84.77** | 41.67 | **89.99** | **69.11** | **77.86** | **90.02** | **90.17** | 93.14 | **42.97** | **85.70** | 64.74 | 87.45 | 86.63 | **88.25** | **87.04** | **68.78** | **90.42** | **59.13** | **79.67** | **78.05** | **78.58** |
| DeepLabv3+ | CE | 94.37 | 90.03 | 42.40 | 82.07 | 70.46 | 75.77 | 93.36 | 88.07 | 90.70 | 36.50 | 86.50 | 67.17 | 86.04 | 90.18 | 87.23 | 85.02 | 68.36 | 88.46 | 57.34 | 84.13 | 78.62 | 78.23 |
|  | CRF-1 | 94.57 | **92.13** | 42.48 | 83.25 | 71.07 | **76.61** | 93.47 | 87.96 | 91.45 | 36.82 | 87.04 | 67.21 | 87.28 | 90.87 | **87.63** | 85.86 | 69.22 | 89.23 | **58.04** | 84.43 | **79.46** | 78.86 |
|  | RMI | **94.97** | 91.57 | **42.93** | **93.72** | **74.84** | 76.23 | **93.68** | **89.09** | **93.59** | **41.99** | **87.63** | **68.79** | **88.23** | **91.33** | 87.12 | **88.62** | **70.24** | **92.00** | 57.77 | 82.53 | 76.60 | **80.16** |

## 5.3 Results on PASCAL VOC 2012 dataset

### 5.3.1 Effectiveness of RMI

RMI is first evaluated on PASCAL VOC 2012 *val* set to demonstrated its effectiveness. The results on *val* set are shown in Tab. 1, With DeepLabv3 and DeepLabv3+ as base models, the RMI can improve the mIoU score by 1.57% and 1.49% on the *val* set respectively. From Tab. 1, we can also see that the reproduced models with RMI outperform the official models with similar settings to ours, and the DeepLabv3 model with RMI can catch the performance of the DeepLabv3+ model without RMI. RMI can also achieve consistent improvements with different models, while the improvements of CRF decrease when the base model is more powerful.

In Tab. 1, RMI outperforms the CRF and affinity field loss, while it has no extra inference steps. Since we downsample the ground truth and probability before we construct the multi-dimensional distributions, RMI only requires a few additional memory. Roughly saying, if we use double-precision floating-point numbers, it needs $2 \times \frac{3 \times 3}{4 \times 4} = 1.125$ times memory than the original memory usage. For a ground truth map with shape $[16, 513, 513, 21]$, theoretically, the additional memory is only about 0.37GB. This is less than the memory that affinity field loss consumed.

We evaluate some models in Tab. 1 on PASCAL VOC 2012 *test* set and the per-class results are shown in Tab. 2. With DeepLabv3 and DeepLabv3+, the improvements are 2.00% and 1.93% respectively. Models trained with RMI show better generalization ability on *test* set than on *val* set.

Some selected qualitative results on *val* set are shown in Fig. 2. Segmentation results of DeepLabv3+&RMI have more accurate boundaries and richer details than results of DeepLabv3+&CE. This demonstrates that RMI can definitely capture the relationship between pixels in an image. Thus predictions of the model with RMI have better visual qualities.

### 5.3.2 Ablation study

In this section, we study the influence of downsampling manner, downsampling factor ($\mathcal{DF}$, the image size after downsampling is the original image size divided by $\mathcal{DF}$), and square region size ($\mathcal{R} \times \mathcal{R}$) on PASCAL VOC 2012 *val* set. The results are shown in Tab. 3.

In Tab. 3a, RMI with average pooling gets the best performance, this may because the average pooling reserves most information after downsampling. Max pooling and nearest interpolation both abandon some points in an image during downsampling. When $\mathcal{DF}$ increase, the performance of RMI with the same region size is likely to be hurt due to the lack of image details. The larger the $\mathcal{DF}$, the smaller the downsampled image size, and the more image details are lost. However, semantic segmentation is a fine-grained classification task, so a large $\mathcal{DF}$ is not a good choice.

In Tab. 3b, we show the influence of the square region size $\mathcal{R} \times \mathcal{R}$, which is also the dimension of the point in the multi-dimension distribution. Here RMI with a large $\mathcal{R}$ and a small $\mathcal{DF}$ is more likely to get better performance. The smaller the downsampling factor $\mathcal{DF}$ and the larger the side length $\mathcal{R}$, the greater the computational resource consumption. The table 3b also shows that RMI with $\mathcal{R} > 1$ gets better result than RMI with $\mathcal{R} = 1$. This further demonstrates that RMI benefits from the relationship between pixels in an image.

There is a trade-off between the performance of RMI and its cost, *i.e.*, GPU memory and floating-point operations. The performance of RMI can be further improved when computational resources are sufficient, *e.g.*, DeepLabv3 with RMI ($\mathcal{DF} = 2, \mathcal{R} = 3$) in Tab. 3a achieve 79.23% on PASCAL VOC 2012 *val* set, which outperforms the 78.85% of improved DeepLabv3+ model reported in [7] with simlilar experimental settings. Limited by our GPU memory, we did not test RMI without downsampling.

Table 3: Influence of different components in RMI. The table 3a shows the effect of the downsampling manner and factor. Avg. is average pooling, Max. is max pooling, and Int. is interpolation. Here we use nearest interpolation for ground truth label and bilinear interpolation for predicted probabilities. The table 3b shows impact of the square region size $\mathcal{R} \times \mathcal{R}$.

| (a) Downsampling manner and factor | | | | |
|---|---|---|---|---|
| | Method | $\mathcal{DF}$ | $\mathcal{R}$ | mIoU (%) |
| | RMI - Int. | 4 | 3 | 77.58 |
| | RMI - Max. | 4 | 3 | 78.62 |
| | RMI - Avg. | 4 | 3 | **78.71** |
| DeepLabv3 | RMI - Avg. | 2 | 3 | **79.23** |
| | RMI - Avg. | 3 | 3 | 79.01 |
| | RMI - Avg. | 4 | 3 | 78.71 |
| | RMI - Avg. | 5 | 3 | 78.50 |
| | RMI - Avg. | 6 | 3 | 78.28 |

| (b) Square region size | | | | |
|---|---|---|---|---|
| | Method | $\mathcal{DF}$ | $\mathcal{R}$ | mIoU (%) |
| | RMI - Avg. | 4 | 1 | 78.25 |
| | RMI - Avg. | 4 | 2 | 78.52 |
| | RMI - Avg. | 4 | 3 | **78.71** |
| DeepLabv3 | RMI - Avg. | 4 | 4 | 78.65 |
| | RMI - Avg. | 6 | 2 | 78.06 |
| | RMI - Avg. | 6 | 3 | 78.28 |
| | RMI - Avg. | 6 | 4 | **78.33** |
| | RMI - Avg. | 6 | 5 | 78.26 |

## 5.4 Results on CamVid dataset

In this section, we evaluate RMI on CamVid dataset to further demonstrate its general applicability. The results are shown in Tab. 4. With DeepLabv3 and DeepLabv3+ as base models, RMI can improve the mIoU score by 2.61% and 2.24% on CamVid *test* set respectively.

From the Tab. 4, we can see that affinity field loss works better on CamVid dataset than on PASCAL VOC 2012 dataset, while CRF behaves oppositely. This suggests that these two methods may not be widely applicable enough. On the contrary, RMI achieves substantial improvement on CamVid dataset as well as PASCAL VOC 2012 dataset. This verifies the broad applicability of the RMI loss.

Table 4: Per-class results on CamVid *test* set. When applying RMI, we employ $\mathcal{DF} = 4$ and $\mathcal{R} = 3$. We choose the best mIoU score from results of CRF with steps 1, 5, and 10.

| | Method | sky | building | pole | road | pavement | tree | sign symbol | fence | car | pedestrian | bicyclist | mIoU (%) |
|---|---|---|---|---|---|---|---|---|---|---|---|---|---|
| DeepLabv3 | CE | 88.06 | 79.03 | 10.12 | 91.32 | 73.21 | 72.65 | 41.04 | 39.52 | 79.35 | 42.43 | 54.51 | 58.50 |
| | CE & CRF | **90.64** | **80.48** | 4.70 | 91.49 | 73.64 | **74.35** | 39.93 | 41.98 | 80.45 | 40.50 | 54.20 | 58.59 |
| | Affinity | 88.23 | 78.85 | 11.54 | **92.77** | **76.50** | 71.96 | 45.41 | 39.27 | **82.65** | 43.26 | 54.14 | 59.60 |
| | RMI | 89.29 | 79.89 | **11.64** | 92.04 | 75.62 | 73.39 | **48.02** | **43.25** | 82.01 | **46.74** | **59.67** | **61.11** |
| DeepLabv3+ | CE | 90.86 | 80.28 | 27.62 | 93.60 | 78.57 | 74.35 | 44.51 | 38.86 | 84.35 | 51.80 | 57.09 | 62.82 |
| | CE & CRF | **91.88** | 80.92 | 15.98 | 93.20 | 78.14 | **74.89** | 42.14 | 39.72 | 84.50 | 40.74 | 56.04 | 61.68 |
| | Affinity | 90.23 | **81.03** | **29.83** | 93.93 | 80.86 | 74.75 | 47.21 | 41.50 | **85.12** | 54.97 | 60.92 | 64.42 |
| | RMI | 90.91 | 80.50 | 28.68 | **94.16** | **81.28** | 73.77 | **52.08** | **42.72** | 83.22 | **57.75** | **62.57** | **65.06** |

# 6 Conclusion

In this work, we develop a region mutual information (RMI) loss to model the relationship between pixels in an image. RMI uses one pixel and its neighbour pixels to represent this pixel. In this case, we get a multi-dimensional point constructed by a small region in an image, and the image is cast into a multi-dimensional distribution of these high-dimensional points. Then the prediction of the segmentation model and ground truth can achieve high order consistency through maximizing the mutual information (MI) between their multi-dimensional distributions. However, it is hard to calculate the value of the MI directly, so we derive a lower bound of the MI and maximize the lower bound to maximize the actual value of the MI. The idea of RMI is intuitive, and it is also easy to use since it only requires a few additional memory during the training stage. Meanwhile, it needs no changes to the base segmentation model. We experimentally demonstrate that RMI can achieve substantial and consistent improvements in performance on standard benchmark datasets.

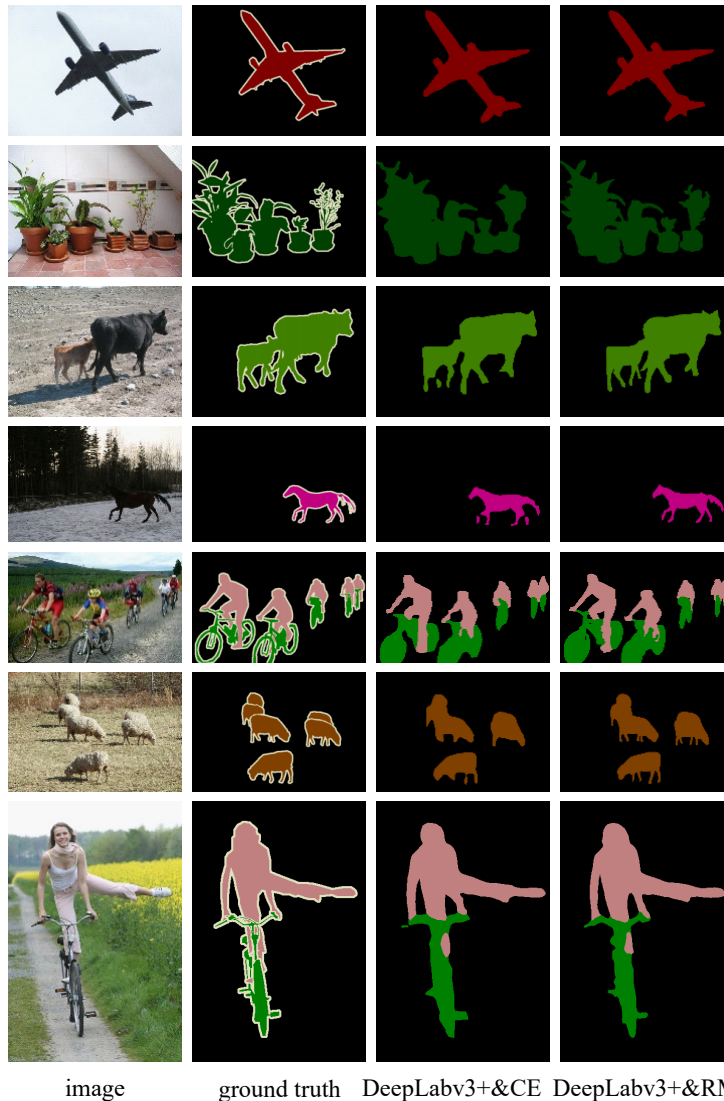

image      ground truth    DeepLabv3+&CE   DeepLabv3+&RMI

Figure 2: Some selected qualitative results on PASCAL VOC 2012 *val* set. Here we use the DeepLab3+ models in Tab. 1. Segmentation results of DeepLabv3+&RMI have richer details than DeepLabv3+&CE, *e.g.*, small bumps of the airplane wing, branches of plants, limbs of cows and sheep, and so on. **Best view in color with 300% zoom**.

## Acknowledgments

This work was supported in part by The National Key Research and Development Program of China (Grant Nos: 2018AAA0101400), in part by The National Nature Science Foundation of China (Grant Nos: 61936006).

## Footnotes

[2]https://github.com/zhanghang1989/PyTorch-Encoding/blob/master/encoding/models/model_zoo.py

[3]http://www.philkr.net/2011/12/01/nips/

[4]https://github.com/lucasb-eyer/pydensecrf

[5]https://github.com/twke18/Adaptive_Affinity_Fields

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
