[Reviews · NeurIPS 2019]

Reviewer 1



Post-rebuttal: Thanks to the authors for a thoughtful rebuttal. If the authors are able to find space for the qualitative results, I believe most of the reviewers' concerns were addressed. I am however wary of adding additional experiments from Pascal 2012 test, even if they were technically done before the submission deadline. The motivation of this approach is that per-pixel losses don't explicitly capture the structural differences between the shapes of predictions in ground truth. By comparing the mutual information between local patches in the prediction and ground truth, the authors hope to encourage higher order similarities. Ultimately the results on Pascal 2012 and CamVid show that such an approach can lead to small but significant improvements in IoU over prior approaches based on CRFs and Pixel Affinities, without any added cost during inference. Along the way they show how to approximately minimize this region mutual information based, with the appropriate downsampling to efficiently train the model with this loss. One interesting potential advantage of this approach is that since the loss is based on aggregate statistics over small regions, it may be more robust to small labeling errors. It would be interesting to see if this is the case by artificially perturbing the training data in various ways (coarser polygons, random shifts between GT/image, etc). Along these lines, it would also be interesting to see the influence of different weight factors (\lambda) between the cross entropy and region mutual information. In particular, I'd be curious if this loss can work without cross entropy at all? Further, it would be interesting to consider a loss that is more sensitive to errors in the shape or higher order errors. For example, something like hausdorff distance between object boundaries might be a suitable metric.

Reviewer 2



1. The main concern with this work is that the benefits of using RMI are not presented well in this paper at all. I am not sure why practitioners would like to use the RMI loss in practice for semantic image segmentation, given only marginal improvements at a more complex computational cost. 2. It is not clear if the assumption (11) holds in experiments for semantic image segmentation. There are no experimental results to support this. 3. I appreciate the details of the equations to explain the proposed RMI loss. However, given the equation 16 and equation 13, how would one to implement it in an efficient way is not very clear after reading the paper. It is also not very clear if the proposed approach would work well with end-to-end training with deep neural networks at all. 4. It is not clear if the proposed loss would give improvements with other base models, and how much this would give compared with the other tricks, such as atrous convolution, various up-sampling modules.

Reviewer 3



- Originality: I think many people have thought of different ways trying to model the relationships between pixels, e.g. (as the paper mentioned) CRF. The current proposal of using region based loss is both novel and intuitive to me. - Quality: The paper is written in a decent quality. The overall style is biased toward being "technical", with clean mathematical definitions and down-to-earth practical details. The only thing missing to me is qualitative results for experiments, which is very essential to me as the paper is focusing on the qualitative difference between a model that takes into account the correlations between pixels vs. a model that does not. - Clarity: Although I haven't check the mathematical formulations, the paper is clear enough in delivering what is the intuition for the solution, and what is the results. - Significance: Judging from the quantitative results, I think the performance improvement is significant enough for a paper. I do find several things lacking: -- The results are reported on the val set for VOC 2012, instead of the test set; -- Qualitative results are lacking; -- It would be nice to include more SOTA comparisons (I am not up to date on this, but is DeepLab v3+ still the state-of-the-art?); -- Another important experiment to analyze is to verify the performance of 1) RMI + CRF; 2) CRF + Affinity; 3) Affinity + RMI. This is to see if RMI has already done the job that CRF/Affinity is doing, or vice versa -- how much of RMI's contribution can be done in CRF/Affinity. -- Another baseline I can think of is to have a "pyramid" of losses, e.g. using the same CE loss or BCE loss, but apply it a pyramid of downsampled predictions. Will that help performance? Will that replace the functionality of RMI?

[Author Response · NeurIPS 2019]

1. **The results on the PASCAL VOC 2012** *test* **set.** They are shown in Tab. 1. With DeepLabv3 and DeepLabv3+, the improvements are 2.00% and 1.93% respectively. Models trained with RMI show better generalization ability on *test* set than on *val* set. The results are supposed to be in Tab.2 in the main paper. However, our submissions are stuck in the official server (too much submissions or some other reasons). You can check the link of DeepLabv3&CE (`http://host.robots.ox.ac.uk:8080/anonymous/ERHL1O.html`) to verify our words – the submission date is 2019-05-23 and we receive the results after 4 days (the time we received the reminder mail). Some earlier attempts on *val* set are also stuck. Furthermore, the links of DeepLabv3&RMI and DeepLabv3+&RMI are `http://host.robots.ox.ac.uk:8080/anonymous/2RBVFL.html` and `http://host.robots.ox.ac.uk:8080/anonymous/SC5YIQ.html`.

Table 1: Per-class results on the PASCAL VOC 2012 *test* set.

| Method | | backg. | aero. | bike | bird | boat | bottle | bus | car | cat | chair | cow | d.table | dog | horse | mbike | person | p.plant | sheep | sofa | train | tv | mIoU (%) |
|---|---|---|---|---|---|---|---|---|---|---|---|---|---|---|---|---|---|---|---|---|---|---|---|
| DeepLabv3 | CE | 94.10 | 79.58 | 41.16 | 84.67 | 67.68 | 75.09 | 87.69 | 87.40 | 92.07 | 39.66 | 83.39 | 69.68 | 86.67 | 87.10 | 86.92 | 84.39 | 65.69 | 86.66 | 57.39 | 75.28 | 75.94 | 76.58 |
| | CRF-5 | 94.60 | 84.28 | **41.83** | 88.00 | 68.81 | 76.56 | 87.69 | 87.90 | **93.79** | 40.35 | 84.92 | **70.26** | **88.84** | **89.22** | 87.39 | 85.73 | 67.45 | 87.95 | 58.80 | 75.31 | 77.38 | 77.96 |
| | RMI | 94.57 | **84.77** | 41.67 | **89.99** | **69.11** | **77.86** | **90.02** | **90.17** | 93.14 | **42.97** | **85.70** | 64.74 | 87.45 | 86.63 | **88.25** | **87.04** | **68.78** | **90.42** | **59.13** | **79.67** | **78.05** | **78.58** |
| DeepLabv3+ | CE | 94.37 | 90.03 | 42.40 | 82.07 | 70.46 | 75.77 | 93.36 | 88.07 | 90.70 | 36.50 | 86.50 | 67.17 | 86.04 | 90.18 | 87.23 | 85.02 | 68.36 | 88.46 | 57.34 | 84.13 | 78.62 | 78.23 |
| | CRF-1 | 94.57 | **92.13** | 42.48 | 83.25 | 71.07 | **76.61** | 93.47 | 87.96 | 91.45 | 36.82 | 87.04 | 67.21 | 87.28 | 90.87 | **87.63** | 85.86 | 69.22 | 89.23 | **58.04** | **84.43** | **79.46** | 78.86 |
| | RMI | **94.97** | 91.57 | **42.93** | **93.72** | **74.84** | 76.23 | **93.68** | **89.09** | **93.59** | **41.99** | **87.63** | **68.79** | **88.23** | **91.33** | 87.12 | **88.62** | **70.24** | **92.00** | 57.77 | 82.53 | 76.60 | **80.16** |

2. **The qualitative results.** They are shown in Fig. 1. The lack of the these results in the main paper is due to the limit of paper length. It is clear that the predictions of DeepLabv3+&RMI have more accurate boundaries and richer details.

Figure 1: The qualitative results on PASCAL VOC 2012 *val* set. **Best view in color with 400% zoom**.

3. **The significance of RMI.** PASCAL VOC dataset is well-studied and DeepLabv3+ is still the best model on it (`http://host.robots.ox.ac.uk:8080/leaderboard/displaylb.php?challengeid=11&compid=6`). The improvement gained by RMI with a well-developed model can definitely demonstrate its effectiveness. ResNet101-DeepLabv3 have a 77.21% mIoU on *val* set [6, Tab. 5], and ResNet101-DeepLabv3+ obtain a 78.85% mIoU (add a decoder based on DeepLabv3) [7, Tab. 3]. In contrast, with the same settings, the best mIoU of ResNet101-DeepLabv3&RMI is 79.09% (Tab. 3a in the main paper). Furthermore, RMI can get consistent improvements with DeepLabv3+. It worth noticed that the top performance of DeepLabv3+ also comes from other paralleled aspects, *e.g.*, small output stride, more powerful backbone network (Xception), COCO&JFT pretraining, and multi-scale&flipping evaluation. They cost more computational resources.

Besides above, RMI provides a practical method to estimate the mutual information through statistics of the data when their corresponding distributions are unknown. Moreover, the idea of RMI is not restricted in semantic segmentation. It can be applied in many other structured output tasks (like some image-to-image tasks), and this is our future work.

4. **Some other questions. Q (Reviewer #1):** The influence of $\lambda$ in Eq.(16). **A:** Following the SSIM index [37], the importance of pixel similarity and structure similarity is equal, so we simply set $\lambda = 0.5$. We further study the influence of $\lambda$ on VOC *val* set with DeepLabv3: {0.1, 0.3, 0.5, 0.7, 0.9} – {77.49, 77.88, **78.71**, 78.50, 77.40}(%, mIoU).

**Q (Reviewer #2):** Assumption (11), training set up, implementation, other tricks, and other base models. **A: (1).** Given assumption (11), we can calculate an approximate value of $I_l(Y; P)$, and the difference between estimated $I_l(Y; P)$ and real value of $I_l(Y; P)$ is restricted in certain range by Theorem 3.1. This assumption is discussed in a more general way in [14]. We are standing on giants' shoulders, check [14] for more details. **(2).** Training set up is clear in Sec 5.1. We keep the set up same as [6, 7] for fair comparisons, and all models in our paper are training end-to-end with a certain loss following the same routine. No fine-tune and pretrained DeepLab models. **(3).** It is discussed in Line.149-162. Code will also be public. **(4).** Atrous convolution and various up-sampling modules are already used in DeepLabv3 and DeepLabv3+ (ASPP and Encoder-Decoder design). Check [5, 6, 7] for more details. **(5).** According to [6, 7], DeepLabv3 and DeepLabv3+ follow two different design principles – the former is plain and the latter is Encoder-Decoder based. Nevertheless, we provide results of PSPNet on VOC *val* set: CE (77.58%), RMI (**78.63%**).

**Q (Reviewer #3):** mixture of RMI/CRF/Affinity, "pyramid" loss. **A: (1).** CRF shows minor improvments when base model is powerful enough (Tab. 1b) and negative effect on CamVid (Tab. 4). Affinity loss shows negative effect on PASCAL VOC. In contrast, the improvements of RMI are consistent, so we think it is unnecessary to mix them up. **(2).** Common downsampling methods may produce the same interpolated result form different regions, so "pyramid" loss is not locality-aware. We examed this idea with DeepLabv3 on VOC *val* set: CE (**77.14%**) , "pyramid" loss (76.11%, simply averaged over scales [0.25, 0.5, 0.75, 1.0]). The "pyramid" loss shows negative effect.

[Meta-Review · NeurIPS 2019]

The reviews were leaning positive, and after the rebuttal, two of the reviewers recommended to accept, while R3 remained at the "marginally below" score. I agree that most of the reviewers comments were addressed in the rebuttal, and concur with the majority vote here. The paper is an incremental, but worthy addition to the evolving semantic segmentation craft. I strongly encourage the authors to incorporate the responses in the rebuttal into the final version, and in particular, to make sure to include some qualitative results.